# CONTEXT-AWARE ATTENTION MODEL FOR COREFERENCE RESOLUTION

## ABSTRACT

Coreference resolution is an important task for gaining more complete understanding about texts by artificial intelligence.The state-of-the-art end-to-end neural coreference model considers all spans in a document as potential mentions and learns to link an antecedent with each possible mention. However, for the verbatim same mentions, the model tends to get similar or even identical representations based on the features, and this leads to wrongful predictions. In this paper, we propose to improve the end-to-end system by building an attention model to reweigh features around different contexts. The proposed model substantially outperforms the state-of-the-art on the English dataset of the CoNLL 2012 Shared Task with 73.45% F1 score on development data and 72.84% F1 score on test data.

## 1 INTRODUCTION

Coreference is one of the most frequent phenomena in English and the other languages. Coreference resolution is a crucial task before artificial intelligent systems capable of fully understanding the human language. Supervised methods, especially the models using neural-network-generated word representations, achieve outstanding performances Clark & Manning (2016); Lee et al. (2017; 2018). However, for similar or identical text units, the problem of wrongfully getting the same coreferences is still puzzling. For example, the following conversation has the sentences, A and B.

A: Yeah,**it's** not far.Through **the S-bahn** here.**I** mean **it's** like twenty minutes.

B: Or something.And so,if **I** do **it**,**I'd** love to have **you** join **me**.**It's** a fancy wedding too.

The pronoun "it's" in the sentence A and the "it's" in the sentence B are obviously referring to the different things. As they are likely to get similar or even the same expression, a false link between them is often predicted. A similar case in sentence B. Due to different lemma and lexeme, the model would not predicate that "I" in A and "you" in B are coreferential. On the opposite, a false coreference between "I" in A and "I" in B would be predicted.

In coreference resolution tasks, words referring each other are called mentions, while a mention could be a common noun, a proper noun or a pronoun. Taking the above example, a coreference system partitions the mentions in a sentence into one coreference chain-("it's", "it's"), and singleton: "the S-bahn" for speaker1. One coreference chain-("I", "I'd","me"), and singleton: "It's" for speaker2 and one coreference chain-("I", "you") between two speakers.

In recent years, several supervised approaches have been proposed to coreference resolution. The work can be categorized into three classes. 1) mention-pair models: A mention pair model is a binary classifier that determines whether a pair of mentions is co-referring or not McCarthy & Lehnert (1995). One of the common limitation of the mention-pair model is that it cannot capture information beyond the mention pair. The information that can be obtained from the two markables to determine their coreferential status is very limited.; 2) entity-mention models: The entity-mention model aims to classify whether an NP (Noun phrase) is coreferent with a preceding cluster Yangy et al. (2004); Culotta et al. (2007); Daumé III & Marcu (2005). This strategy considers the candidates independently. It cannot measure how likely a candidate is the antecedent for a given anaphor, relative to the other candidates; 3) ranking models: Ranking models allows candidate antecedents of a mention to be ranked simultaneously Iida et al. (2003); Denis & Baldridge (2008); Durrett & Klein (2013).

In the above three classes, the ranking models recently obtained the state-of-the-art performance Wiseman et al. (2015; 2016); Clark & Manning (2016). More recently, Lee et al. (2018) proposed the first state-of-the-art end-to-end neural coreference resolution system. They consider all spans as potential mentions and learn distributions over possible antecedents for each. In addition, they use a fully differentiable approximation to higher-order inference to iteratively refine span representations, and the model only uses a minimal set of hand-engineered features (speaker ID, document genre, span distance, span width). This leads to the problem that identical mentions tend to get similar or even identical representations, and further misled coreference resolutions to make mistakes.

In this paper, we demonstrate a novel method utilizing attention mechanism to adaptively exploit features to represent identical mentions with different contexts. Inspired by the recent success of attention mechanism, we focus on this issue and develop a general attention mechanism that learns the importance/weight of each feature based on mention's contexts and then add this information to the end-to-end neural model. The entire model could be trained end-to-end with gradient descent.

The proposed model is evaluated on the CoNLL 2012 Shared Task Pradhan et al. (2012). The results show that the method outperforms the baselines. Meanwhile, we made a statistic of the different features' weights in the attention mechanism. The statistic shows that the feature attention algorithm does help to distinguish the features for identical mentions based on different contexts.

## 2 THE MODEL

### 2.1 TASK

In an end-to-end coreference resolution, the input is a document $D$ with $T$ words along with meta-data, and the output is a set of mention clusters. Let $N$ be the number of possible text spans in $D$. We consider all possible spans up to a predefined maximum width. Denote the start and end indices of a span $i$ in $D$ respectively by $START(i)$ and $END(i)$. For each span $i$ the system needs to assign an antecedent $a_i \in \{\epsilon, 1, ..., i\text{-}1\}$ from all preceding spans or a dummy antecedent $\epsilon$. The dummy antecedent represents two cases: (1) the span $i$ is not an entity mention, or (2) the span $i$ is an entity mention but not coreferential with any previous span. Finally, all spans, that are connected by a set of antecedent predictions, are grouped.

### 2.2 METHODOLOGY

The section elaborates the model. We adopt a similar span representation approach as it in Lee et al. (2018) using bidirectional LSTMs and a headfinding attention. In our approach, we investigate grammatical numbers and use a general attention mechanism reweighing span features based on mention's contexts to generate the new feature representation. Thereafter, the scores of how likely spans could be entity mentions are generated by a feed forward network. We also propose an attention model of antecedent scoring to reweigh pair-wise features and produce the new feature vectors.

**Span Representation** Illustrated in Figure 1,we assume vector representation of a sentence with L words as $\{x_1, x_2, ..., x_L\}$, while $x_t$ denotes the concatenation of fixed pretrained word embeddings and CNN character embeddings Santos & Zadrozny (2014) for t-th word. Then Bidirectional LSTMs Hochreiter & Schmidhuber (1997) are used to encode each $x_t$:

$$\overleftarrow{h_t} = LSTM_{backward}(\overleftarrow{h_{t+1}}, x_t) \tag{1}$$

$$\overrightarrow{h_t} = LSTM_{forward}(\overrightarrow{h_{t+1}}, x_t) \tag{2}$$

$$x_t^* = [\overleftarrow{h_t}, \overrightarrow{h_t}] \tag{3}$$

Then, the model learns a task-specific notion of headedness using the attention mechanism Bahdanau et al. (2014) over words in each span:

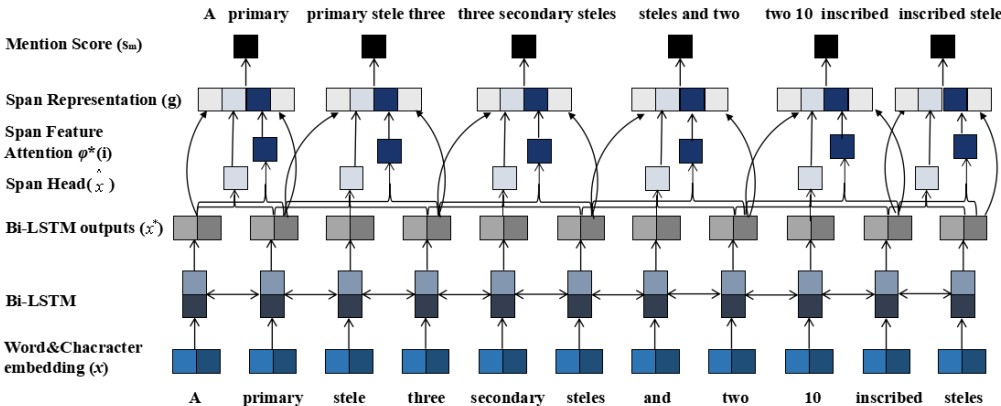

Figure 1: The model of computing the span embedding representations.

$$a_t = w_a \cdot FFNN(x_t^*) \tag{4}$$

$$a_{i,t} = \frac{exp(a_t)}{\sum_{k=START(i)}^{END(i)} exp(a_k)} \tag{5}$$

$$\hat{x}_i = \sum_{t=START(i)}^{END(i)} a_{i,t} \cdot x_t \tag{6}$$

Where $x_i$ is a weighted sum of word vectors in span $i$. FFNN is a feed forward network. Then the final representation $g_i$ of span $i$ was produced by:

$$g_i = [x_{START(i)}^*, x_{END}(i)^*, \hat{x}_i, \varphi(i)] \tag{7}$$

Where in Lee et al. (2018), $\varphi(i)$ is a 20-dimensional feature vector encoding only span width information. In our model, we incorporate one more feature, grammatical numbers. The grammatical numbers are reinforced with the feature attention method by reweighing contextual features. A new feature vector $\varphi^*$. is generated by the method.

**Feature Attention** Shown in Figure 2, we use a general attention mechanism that learns the importance/weight of each feature based on contexts. Suppose the initial feature vectors is $\varphi = [\varphi_1, \varphi_2, \dots, \varphi_V]$, where $\varphi_i \in R^{20}$ indicates the i-th feature and $x_u$ is the contexts vectors generated by Bi-LSTM. Then the model learns the weight of each feature based on the contexts, and generate the new feature vectors $\varphi^*$:

$$a_j' = w_a \cdot FFNN(\varphi_j f(x_u)) \tag{8}$$

$$a_{j,u}' = \frac{exp(a_j')}{\sum_{v=1}^{V} exp(a_v')} \tag{9}$$

$$\varphi^* = \oplus_{j=1}^{V} a_{j,u}' \cdot \varphi_j \tag{10}$$

Where $\oplus$ is the concatenate operation and $i$ is a linear function to map $x_u$ to the same dimension with the feature vector. $\varphi^*$ is the new reweighed feature vectors and $a_{j,u}'$ is the weight of each feature based on the contexts.

**Mention Scoring** The new span representation is encoded to a FFNN to compute the mention score measuring if it is an entity mention, where FA is the feature attention algorithm:

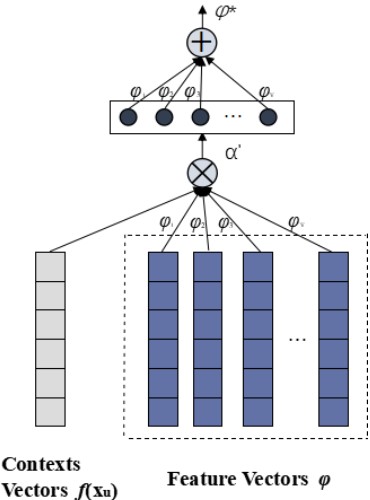

Figure 2: The Feature Attention model. The model learns to weigh each feature based on contexts.

$$\varphi^*(i) = FA(\varphi(i), x_i^*) \tag{11}$$

$$g_i = [x_{START(i)}^*, x_{END(i)}^*, \widehat{x_i}, \varphi^*(i)] \tag{12}$$

$$s_m(i) = w_m \cdot FFNN_m(g_i) \tag{13}$$

**Antecedent Scoring** For antecedent scoring, we use the same features (speaker, genre, distance) as Lee et al. (2018). And incorporate our feature attention algorithm to the features $\varphi(i, j)$:

$$\varphi^*(i, j) = FA(\varphi(i, j), x_i^* \oplus x_j^*) \tag{14}$$

$$s_a(i, j) = w_a \cdot FFNN_a(g_i, g_j, g_i \circ g_j, \varphi^*(i, j)) \tag{15}$$

**Coreference Score** The final coreference score of span $i$ and $j$ indicates that (1) whether span $i$ is a mention, (2) whether span $j$ is a mention, and (3) whether $j$ is an antecedent of $i$ :

$$s(i, j) = \begin{cases} 0, & j = \varepsilon \\ s_m(i) + s_m j + s_a(i, j), & j \neq \varepsilon \end{cases} \tag{16}$$

**Loss Regression** we use the same loss regression as Lee et al. (2018), where GOLD(i) is the set of spans in the gold cluster containing span $i$ :

$$L = -log \prod_{i=1}^{N} \sum_{y' \in Y(i) \cap GOLD(i)} p(y') \tag{17}$$

## 3 EXPERIMENTS AND RESULTS

### 3.1 EXPERIMENTAL SETUP

**Dataset and baseline.** The experiments are conducted on the English subset of the CONLL2012 Shared Task data Pradhan et al. (2012) and evaluated on three standard metrics: MUC Wiseman et al. (2016), B3 Bagga & Baldwin (1998), and CEAF$\varphi$4 Luo (2005). We report precision, recall, F1 for each metric and the average F1 as the final CoNLL score. We use the baseline below:

Table 1: Results on CoNLL 2012 English development set.

| System | MUC | | | B | | | C | | | |
|---|---|---|---|---|---|---|---|---|---|---|
| | P | R | F | P | R | F | P | R | F | Avg F1 |
| Lee et al. (2018) | 82.06 | 77.85 | 79.90 | 74.15 | 68.86 | 71.41 | 69.27 | 66.47 | 67.84 | 73.05 |
| +pair-wise FA | 81.53 | **78.54** | 80.01 | 73.06 | 70.03 | 71.77 | 69.29 | 67.27 | 68.26 | 73.36 |
| +grammatical numbers | 81.65 | 78.43 | 80.00 | 73.97 | 69.60 | 71.72 | 69.05 | **67.60** | 68.32 | 73.35 |
| +span FA | **82.07** | 78.30 | **80.10** | **74.20** | **69.63** | **71.84** | **69.54** | 67.23 | **68.37** | **73.45** |

Björkelund & Kuhn 2014: The system obtains significant improvements over the baseline by modifying LaSO to delay updates until the end of each instance.

Wiseman et al. 2015: The system learns distinct feature representations for anaphoricity detection and antecedent ranking, encouraged by pre-training on a pair of corresponding subtasks.

Wiseman et al. 2016: The system proposes to use recurrent neural networks (RNNs) to learn latent, global representations of entity clusters directly from their mentions.

Clark & Manning 2016: The system presents a neural network based coreference system that produces high-dimensional vector representations for pairs of coreference clusters to learn when combining clusters is desirable.

Lee et al. 2017: The first state-of-the-art end-to-end neural coreference resolution system and only uses a minimal set of hand-craft features.

Lee et al. 2018:The improved model for Lee et al. (2017) using the antecedent distribution from a span-ranking architecture as an attention mechanism to iteratively refine span representations.

**Hyperparameters**. we follow the same hyperparameters as in Lee et al. (2018): Using GloVe word embeddings Pennington et al. (2014) with a window size of 2 for the head word embeddings and a window size of 10 for the LSTM inputs and embedding representations from a language model Peters et al. (2018) as the input to the LSTMs (ELMo in the results). The maximum span width was 30 words and only consider 50 antecedents. $\lambda = 0.4$ is used for span pruning. The model is trained up to 150 epoch with early stopping.

## 3.2 RESULTS AND DISCUSSION

**Contrast Experiments** We first run the Lee2018 system[1] on the development data and test data of CONLL2012 Shared Task. To train our model, we employ the same hyper-parameters as reported in Lee et al. (2018). Then the model is trained and compared with the Lee2018 system. We compare the model with the baseline system with each improvement : (1) Pair-wise FA (Feature Attention): incorporating feature attention algorithm to the pair-wise features $\varphi(i,j)$; (2) Grammatical numbers (GN): incorporating grammatical numbers to span features $\varphi(i)$; (3) Span FA(Feature Attention): using feature attention algorithm to encode span feature $\varphi(i)$.

**Comparison and Analysis** To investigate the importance of each component in the proposed model, we report the results of each part in Table 1 and Table 2, comparing with Lee2018 system.

Pair-wise Feature Attention: The performance increases by 0.31 F1 on dev dataset and 0.17 on test with the Pair-wise FA. The improvement of the precision shows that the false-positive links for the identical or similar mentions (not coreferent) are decreased. In the baseline model, when mention $t$ is predicated "not coreferent" to the mention $u$, the other mentions identical to $t$ will be likely to be predicated "not coreferent" to the mention $u$ too. But the significant improvements from the recall indicates that such false negative links are substantially decreased by the model.

Grammatical numbers: The addition of a new span feature, grammatical numbers, helps little with the system. This indicates that the grammatical information extraction of the whole system is relatively sufficient. So the GN does not help much.

---

[1]https://github.com/kentonl/e2e-coref/

Table 2: Results on the CoNLL 2012 English test set.

| System | MUC | | | B | | | C | | | |
|---|---|---|---|---|---|---|---|---|---|---|
| | P | R | F | P | R | F | P | R | F | Avg F1 |
| Lee et al. (2018) | **81.8** | 78.54 | 80.14 | **73.2** | 68.15 | 70.58 | 68.51 | 65.94 | 67.20 | 72.64 |
| +pair-wise F A | 81.71 | 78.84 | 80.24 | 72.82 | 68.60 | **70.67** | 68.82 | 66.26 | 67.51 | 72.81 |
| +grammatical numbers | 81.30 | 79.13 | 80.20 | 72.16 | **69.11** | 70.60 | 68.69 | **66.36** | 67.51 | 72.77 |
| +span F A | 81.42 | **79.20** | **80.29** | 72.44 | 68.94 | 70.65 | **69.12** | 66.08 | **67.57** | **72.84** |

Table 3: Overall performance on CoNLL 2012 English test set.

| System | MUC | | | B | | | C | | | |
|---|---|---|---|---|---|---|---|---|---|---|
| | P | R | F | P | R | F | P | R | F | Avg F1 |
| B&K (2014) | 74.30 | 67.46 | 70.72 | 62.71 | 54.96 | 58.58 | 59.40 | 52.27 | 55.61 | 61.63 |
| Wiseman et al. (2015) | 76.23 | 69.31 | 72.6 | 66.07 | 55.83 | 60.52 | 59.41 | 54.88 | 57.05 | 63.39 |
| Wiseman et al. (2016) | 77.49 | 69.75 | 73.42 | 66.83 | 56.95 | 61.50 | 62.14 | 53.85 | 57.70 | 64.21 |
| Clark & Manning (2016) | 79.91 | 69.30 | 74.23 | 71.01 | 56.53 | 62.95 | 63.84 | 54.33 | 58.70 | 65.29 |
| Lee et al. (2017) | 78.40 | 73.40 | 75.80 | 68.60 | 61.80 | 65.00 | 62.70 | 59.00 | 60.80 | 67.20 |
| Lee et al. (2018) | **81.80** | 78.54 | 80.14 | **73.20** | 68.15 | 70.58 | 68.51 | 65.94 | 67.20 | 72.64 |
| This work | 81.42 | **79.20** | **80.29** | 72.44 | **68.94** | 70.65 | **69.12** | 66.08 | **67.57** | **72.84** |

Span Feature Attention: Finally, we consider the span feature attention. We see a contribution of 0.1 F1 on dev dataset and 0.13 on test. The results show that the span FA does help to improve the system, especially in the mention detection subtask.The contribution in mention detection subtask is further discussed in Section Mention Detection Subtask.

**Overall Performance** Overall performance comparing with the other state-of-the-art systems are shown in Table 3. We could see a significant improvement on average F1 scores over the previous work, and the highest F1 scores on all three metrics especially in CEAF$\varphi$4 metric. And the most significant gains come fron improvements of recall. Such improvements indicate that the applying of Feature Attention algorithm does help to distinguish the features for identical spans and mentions based on different contexts.That means the weights of different features are useful for making coreference decisions.

**Mention Detection Subtask** To further understand the utility of Feature Attention algorithm for mention detection subtask, we list the mention detection performance in Table 4. Compared wirh Lee2018, the performance increases by 0.33 F1 on dev dataset and 0.31 on test. And the results show that the model indeed performs better in the recall scores. In the baseline model, when there is a span not predicated to be a mention, the other identical spans will unlikely to be detected as mentions, either. However, in our model, such false negative links are decreased, being benefit from the span FA algorithm that reweighs the features to distinguish identical spans with different representations based on different contexts.

## 3.3 FEATURE WEIGHTS

To gain a further insight about how identical terms' representations could be distinguished by the attention mechanism, the feature attention weights are investigated. For example, span feature attention weights are shown in Figure 3. For the three "it's" in the figure, we can observe that the first two lines of "it's" that are coreferent gain similar weights for the two features: higher weight in span width than grammatical number, but the third "it's" not coreferent gains the opposite results.This indicates that the identical spans which have the same features will have the different span feature attention weights depending on their contexts through the span feature attention and finally improves the performance.

Table 4: Mention Detection Results on CoNLL 2012 English data

| System | CoNLL-2012 Dev Set | | | CoNLL-2012 Test Set | | |
|---|---|---|---|---|---|---|
| | P | R | F | P | R | F |
| Lee et al. (2018) | 86.71 | 82.49 | 84.55 | 86.62 | 83.22 | 84.89 |
| Our work | **86.78** | **83.05** | **84.88** | **86.64** | **83.80** | **85.20** |

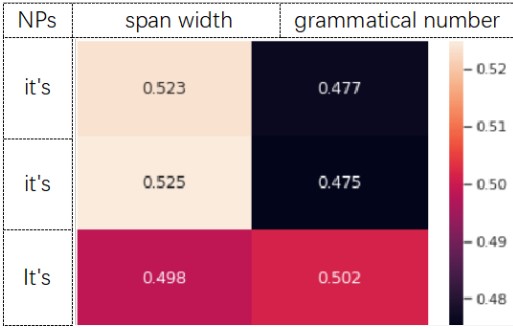

Figure 3: Example of span feature attention weights in different features. The rows show the span feature attention weights of features of each "it's".

## 4 CONCLUSION

Identical mentions impose difficulties on the current methods of coreference resolution as they tend to get similar or even identical representations. The problem may directly lead a coreference resolution to wrong predictions. In the paper, we focus on this issue and develop an attention model named Feature Attention to adaptively exploit features in order to represent identical mentions with consideration of different contexts. The results show that our model with the Feature Attention algorithm performed reasonably well in coreference resolution, which is evaluated on the CoNLL-2012 Shared Task in English.

## ACKNOWLEDGMENTS

This work has been supported by the National Key Research and Development Program of China (2018YFC0910404); National Natural Science Foundation of China (61772409); Ministry of Education-Research Foundation of China Mobile Communication Corp (MCM20160404); the consulting research project of the Chinese Academy of Engineering (The Online and Offline Mixed Educational Service System for "The Belt and Road" Training in MOOC China); Project of China Knowledge Centre for Engineering Science and Technology; the innovation team from the Ministry of Education(IRT₋17R86); and the Innovative Research Group of the National Natural Science Foundation of China (61721002).Professor Chen Li' s Recruitment Program for Young Professionals of "the Thousand Talents Plan"

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
