# OpenReview forum: "Context-aware Attention Model for Coreference Resolution"
_ICLR.cc/2020/Conference — Reject_

### Official Review · AnonReviewer3 · 2019-10-22
**Official Blind Review #3**

**Rating:** 1

**Review:**

This paper proposes to use an extra feature (grammatical number) for context-aware coreference resolution and an attention-based weighting mechanism. The approach proposed is built on top of a recent well performing model by Lee et al. The improvement is rather minor in my view: 72.64 to 72.84 in the test set.

There is not much in the paper to review. I don't think the one extra feature warrants a paper at a top conference. The weighting mechanism over the features is also unclear to me why it benefits from attention. Couldn't we just learn the weights using another layer? It could be context dependent if desired.

It is also incorrect to criticise Lee et al. (2018) that they would give the same representation to the same mention every time. Their model is context dependent as they use a BiLSTM over the sentence. Of course the same mentions are likely to get similar representations, but this is desirable.

**Experience Assessment:**

I have read many papers in this area.

**Review Assessment: Checking Correctness Of Derivations And Theory:**

N/A

**Review Assessment: Checking Correctness Of Experiments:**

I assessed the sensibility of the experiments.

**Review Assessment: Thoroughness In Paper Reading:**

N/A

---

### Official Review · AnonReviewer2 · 2019-10-23
**Official Blind Review #2**

**Rating:** 1

**Review:**

This paper extends the neural coreference resolution model in Lee et al. (2018) by 1) introducing an additional mention-level feature (grammatical numbers), and 2) letting the mention/pair scoring functions attend over multiple mention-level features. The proposed model achieves marginal improvement (0.2 avg. F1 points) over Lee et al., 2018, on the CoNLL 2012 English test set.

I recommend rejection for this paper due to the following reasons:
- The technical contribution is very incremental (introducing one more features, and adding an attention layer over the feature vectors).
- The experiment results aren't strong enough. And the experiments are done on only one dataset.
- I am not convinced that adding the grammatical numbers features and the attention mechanism makes the model more context-aware.

Other suggestions:
- The citation format seems weird through out the paper.
- Table 1 and 3 look like ablation results. It might be less confusing if it's presented as "Full system: xx%; without pairwise FA: yy%; without grammatical numbers zz% ...".
- Equation 8 - 10 are quite confusing. What is f(x)? How large is V? What is u? etc.
- Please define/explain the "grammatical numbers" feature when it's introduced in Section 2.2.

**Experience Assessment:**

I have published in this field for several years.

**Review Assessment: Checking Correctness Of Derivations And Theory:**

I assessed the sensibility of the derivations and theory.

**Review Assessment: Checking Correctness Of Experiments:**

I assessed the sensibility of the experiments.

**Review Assessment: Thoroughness In Paper Reading:**

I read the paper at least twice and used my best judgement in assessing the paper.

---

### Official Review · AnonReviewer1 · 2019-10-23
**Official Blind Review #1**

**Rating:** 1

**Review:**

This paper unfortunately violates the blind-review policy: its acknowledgement exposes the authors. I thus support desk rejection.


**Experience Assessment:**

I have published one or two papers in this area.

**Review Assessment: Checking Correctness Of Derivations And Theory:**

I assessed the sensibility of the derivations and theory.

**Review Assessment: Checking Correctness Of Experiments:**

I assessed the sensibility of the experiments.

**Review Assessment: Thoroughness In Paper Reading:**

I read the paper thoroughly.

---

### Decision · Program_Chairs · 2019-12-19

**Decision:**

Reject

**Comment:**

Main content:

Blind review #2 summarizes it well:

This paper extends the neural coreference resolution model in Lee et al. (2018) by 1) introducing an additional mention-level feature (grammatical numbers), and 2) letting the mention/pair scoring functions attend over multiple mention-level features. The proposed model achieves marginal improvement (0.2 avg. F1 points) over Lee et al., 2018, on the CoNLL 2012 English test set.

--

Discussion:

All reviewers rejected.

--

Recommendation and justification:

The paper must be rejected due to its violation of blind submission (the authors reveal themselves in the Acknowledgments).

For information, blind review #2 also summarized well the following justifications for rejection:

I recommend rejection for this paper due to the following reasons:
- The technical contribution is very incremental (introducing one more features, and adding an attention layer over the feature vectors).
- The experiment results aren't strong enough. And the experiments are done on only one dataset.
- I am not convinced that adding the grammatical numbers features and the attention mechanism makes the model more context-aware.